# Identification, Pathogenicity, and Genetic Diversity of *Fusarium* spp. Associated with Maize Sheath Rot in Heilongjiang Province, China

**DOI:** 10.3390/ijms231810821

**Published:** 2022-09-16

**Authors:** Xilang Yang, Xi Xu, Shuo Wang, Li Zhang, Guijin Shen, Haolin Teng, Chunbo Yang, Chunru Song, Wensheng Xiang, Xiangjing Wang, Junwei Zhao

**Affiliations:** 1Key Laboratory of Agricultural Microbiology of Heilongjiang Province, School of Life Sciences, Northeast Agricultural University, No. 600 Changjiang Road, Harbin 150030, China; 2State Key Laboratory for Biology of Plant Diseases and Insect Pests, Institute of Plant Protection, Chinese Academy of Agricultural Sciences, Beijing 100097, China

**Keywords:** maize, genetic diversity, *Fusarium* spp., haplotype analysis, pathogenicity test

## Abstract

Maize sheath rot is a prevalent maize disease in China. From 2020 to 2021, symptomatic samples were collected from the main maize-growing regions of Heilongjiang province. To clarify the population and genetic diversity, as well as the virulence of pathogens responsible for maize sheath rot, a total of 132 *Fusarium* isolates were obtained and used for follow-up studies. Ten *Fusarium* species were identified based on morphological characteristics, and phylogenetic analysis was conducted using the TEF-1α gene sequences, including *F. verticillioides* (50.00%), *F. subglutinans* (18.94%), the *Fusarium incarnatum-equiseti* species complex (14.39%), *F. temperatum* (5.30%), *F. acuminatum* (3.03%), *F. solani* (2.27%), *F. sporotrichioides* (2.27%), *F. tricinctum* (1.52%), *F. asiaticum* (1.52%), and *F. proliferatum* (0.76%). All 10 *Fusarium* species could produce oval-to-annular lesions on maize sheath, and the lesions were grayish yellow to dark brown in the center and surrounded by a dark gray-to-dark brown halo. Of these, *F. tricinctum* and *F. proliferatum* showed significantly higher virulence than the other *Fusarium* species. In addition, haplotype analysis based on the concatenated sequences of the ITS and TEF-1a genes showed that 99 *Fusarium* isolates which belonged to the *Fusarium fujikuroi* species complex—consisting of *F. verticillioides* isolates, *F. subglutinans* isolates, *F. temperatum* isolates, and *F. proliferatum* isolates—could be grouped into 10 haplotypes, including 5 shared haplotypes (Haps 1, 2, 4, 5, and 6) and 5 private haplotypes (Haps 3, 7, 8, 9, and 10). Furthermore, the *F. verticillioides* clade in the haplotype network was radial with the center of Hap 2, suggesting that population expansion occurred. This research showed that *Fusarium* species associated with maize sheath rot in Heilongjiang province are more diverse than previously reported, and this is the first time that *F. subglutinans*, *F. temperatum*, *F. solani*, *F. sporotrichioides*, *F. tricinctum*, and *F. acuminatum* have been confirmed as the causal agents of maize sheath rot in Heilongjiang province.

## 1. Introduction

Maize is one of the three major food crops worldwide and the main raw feed material. It is also widely used in medicine, energy, and other fields. In China, maize is an important food crop, and the planting area was 41,264 million hectares in 2020 [1]. Meanwhile, maize is also the most important crop in Northeast China, accounting for more than 50% of the planting area of crops [2]. Maize sheath rot was first recorded in 2008 in Northeast China and has been reported to cause varying degrees of maize yield loss [3,4,5,6,7]. In addition, studies also showed that maize sheath rot has occurred in more than 11 provinces of China, including Liaoning, Jilin, Heilongjiang, Hebei, Shandong, Shanxi, Jiangsu, Hunan, Sichuan, Shaanxi, and Gansu, as well as the Ningxia Hui Autonomous Region [8]. The most prevalent pathogens of maize sheath rot are *Fusarium* species, and the diversity of *Fusarium* species causing maize sheath rot is gradually increasing [8].

The genus *Fusarium* was first described by Link in 1809, based on the distinctive banana- or canoe-shaped conidia [9]. Many *Fusarium* species could infect maize and cause a series of diseases, and could also contaminate agricultural products through mycotoxins [10]. As far as we know, maize production can be affected by several diseases caused by *Fusarium* species, such as maize leaf blight, maize stalk rot, maize ear rot, maize seedling blight, maize root rot, maize crown rot, maize storage molds, and maize sheath rot [4,11,12,13,14,15,16]. The *Fusarium fujikuroi* species complex (FFSC) was first established in 1925 by Wollenweber et al. [17] and contained more than 60 phylogenetic species, including *F. proliferatum*, *F. subglutinans*, *F. verticillioides*, *F. temperatum*, *F. fujikuroi*, etc., which were well-known for their ability to cause plant diseases. Members of the FFSC could infect many plants, including mango, *Amaranthus cruentus*, rice, soybean, *Lilium lancifolium*, banana, melon, pineapple, sugarcane, maize, peach, sorghum, *Clivia miniate*, etc. [18,19,20,21,22,23,24,25,26,27,28,29,30,31,32].

*F. proliferatum* was first reported as the causal agent of maize sheath rot in Northeast China [3]. Additionally, *F. graminearum*, *F. verticillioides*, *F. fujikuroi*, *F. meridionale*, *F. asiaticum*, and *F. equiseti* have also been identified as causal agents of maize sheath rot [8]. Symptomatic maize leaf sheaths initially showed small yellow, grayish-brown, brown, or black spots, and then, the lesions gradually expanded into circular, oval, or irregular spots with dark reddish or brown margins. Multiple spots converged to form irregular lesions in the late stage and extended to the whole leaf sheath, resulting in rot of the leaf sheath [3,5,8]. However, current studies on maize sheath rot mainly focus on the identification and pathogenicity of pathogens, and the diversity of *Fusarium* species associated with maize sheath rot is still unclear. Haplotype analysis was used to analyze population genetic data, visualize genealogical relationships, and provide information about the biogeography and history of populations [33]. Identifying the composition of specific haplotypes in particular individuals is very important to understand the phenotype, genetic diversity, and recombination mechanism. In addition, haplotype analysis could also play an important role in mycology research [34].

So far, there have been few studies on *Fusarium* causing maize sheath rot worldwide, and the phylogenetic relationship, phenotypic characteristics, and haplotypes of pathogens are still undefined. Therefore, the purposes of this study were to: (i) isolate and identify pathogens associated with maize sheath rot in different maize-growing regions of Heilongjiang province, (ii) evaluate the pathogenicity of the *Fusarium* isolates obtained in this study, and (iii) determine the genetic diversity of the FFSC based on haplotype analysis.

## 2. Results

### 2.1. Isolation and Identification of Fusarium Isolates

In 2020, 53 samples of maize sheath rot were collected from six maize-growing areas of Heilongjiang Province (including Qitaihe city, Wuchang city, Suihua city, Harbin city, Qiqihar city, and Shuangyashan city); In 2021, 45 samples were collected from maize-growing areas of Heilongjiang Province (including Qitaihe city, Jiamusi city, Suihua city, Harbin city, Qiqihar city, and Shuangyashan city). A total of 98 symptomatic maize leaf sheath samples were collected from seven cities in the main maize-growing areas of Heilongjiang province during 2020–2021. Additionally, Shuangyashan city had the highest frequency of isolation (Figure 1).

A total of 132 *Fusarium* isolates obtained from symptomatic maize leaf sheaths, collected from seven cities in the main maize-growing areas of Heilongjiang province (Appendix A), were tentatively classified into 10 groups according to their morphological characteristics (Table 1, Figure 2 and Figure 3), including *F. verticillioides*, *F. subglutinans*, *F. proliferatum*, *F. temperatum*, *F. tricinctum*, *F. solani*, *F. sporotrichioides*, *F. asiaticum*, *F. acuminatum*, and the *Fusarium incarnatum-equiseti* species complex (FIESC).

Of the isolates, 66 were identified as the *F. verticillioides* morphological group, which could produce white to greyish-purple colonies with dark yellow to purple–gray reverse. The macroconidia were slightly falcate-to-almost straight, with three to four transverse septa, and in a size range of 16.6–23.4 × 2.3–3.4 μm (*n* = 30).

Twenty-five isolates initially produced yellow mycelia, and then, turned lavender; these were identified as the *F. subglutinans* morphological group. The macroconidia were slightly falcate and septate, with three to five transverse septa, in a size range of 13.6–21.1 × 1.6–2.3 μm (*n* = 30).

Nineteen isolates forming white to light beige with loosely floccose mycelia and producing light yellow pigmentation were classified into the FIESC morphological group. The macroconidia were falcate, with three to six transverse septa and in a size range of 16–18 × 1.5–3 μm (*n* = 30).

Seven isolates forming white to peach-colored mycelia were identified as the *F. temperatum* morphological group. Typical macroconidia were falcate, with three to five septa, and in a size range of 13.6–23.1 × 1.6–3.3 μm (*n* = 30).

Four isolates forming abundant and partly carmine mycelia were identified as the *F. acuminatum* morphological group. The macroconidia were slender, equilaterally curved, septate with three to five transverse septa, and in a size range of 13.2–24.2 × 1.9–3.7 μm (*n* = 30). Three isolates forming dense and white mycelia were classified into the *F. solani* morphological group. The macroconidia were abundant and sickle-shaped, with two to four transverse septa, and in a size range of 16–26 × 1.5–2.8 μm (*n* = 30).

Three isolates initially produced white mycelia, and then, turned pink; these were identified as the *F. sporotrichioides* morphological group. The macroconidia were sickle-shaped, with three to five septa, moderately curved-to-straight, and measured 12.5–24.3 × 3.1–4.2 μm (*n* = 30).

Two isolates forming white-to-canary yellow, abundant, and dense mycelia were classified into the *F. tricinctum* morphological group. The macroconidia were slightly curved-to-falcate, with three to five septa, and in a size range of 12.5–26.0 × 1.5–3.0 µm (*n* = 30).

Two isolates forming white mycelia with pink pigmentation were classified into the *F. asiaticum* morphological group. The macroconidia were falcate, with four to eight transverse septa, and in a size range of 13–36 × 1.6–3.8 μm (*n* = 30).

One isolate forming white mycelia with dark violet pigmentation was identified as the *F. proliferatum* morphological group. The macroconidia were sparse, slender, with three to four septa, and measured 13.6–22.3 × 3.1–4.2 μm (*n* = 30).

### 2.2. Phylogenetic Analysis of PCR-Generated DNA Sequences

For further molecular verification, a maximum likelihood tree based on the TEF-1α gene sequences of 132 *Fusarium* isolates and 11 reference strains (Appendix A) was constructed, and these isolates were also grouped into 10 species, including *F. verticillioides* (*n* = 66), *F. subglutinans* (*n* = 25), the FIESC (*n* = 19), *F. temperatum* (*n* = 7), *F. acuminatum* (*n* = 4), *F. solani* (*n* = 3), *F. sporotrichioides* (*n* = 3), *F. tricinctum* (*n* = 2), *F. asiaticum* (*n* = 2), and *F. proliferatum* (*n* = 1) (Figure 4).

### 2.3. Pathogenicity Tests

All *Fusarium* isolates could cause maize sheath rot in this study (Table 2 and Appendix A). The symptoms caused by the 10 *Fusarium* species were similar between species and initially formed grayish-white spots, the spots were gradually enlarged and surrounded by dark brown halos, and the center of the lesions turned to grayish yellow or dark gray with time (Figure 5). The symptoms of maize sheath rot presented in this study were similar to the field symptoms (Figure 5), while no symptoms appeared in the control group (Figure 5). To fulfill Koch’s postulates, the *Fusarium* isolates were all re-isolated from the symptomatic maize leaf sheaths and confirmed according to morphological and molecular methods, whereas no Fusarium isolates were obtained from the control group. The disease incidence and disease index caused by *Fusarium* species were 100% and 11.1–100 (Table 2), respectively. Of which *F. tricinctum* and *F. proliferatum* showed significantly higher virulence than the other *Fusarium* species, with an average disease index of 94.4; this was followed by *F. asiaticum*, *F. acuminatum*, and *F. solani*, which showed similar disease indexes of 73.6, 68.1, and 61.1, respectively. Moreover, *F. sporotrichioides*, *F. temperatum*, *F. verticillioides*, *F. subglutinans*, and the FIESC exhibited somewhat lower virulence, with average disease indexes ranging from 42.3 to 58.3.

### 2.4. Haplotype Network

The haplotype network was constructed based on the concatenated sequences of rDNA ITS and TEF-1α genes, and 10 haplotypes were identified among the 99 FFSC isolates, of which five haplotypes (Haps 1, 2, 4, 5, and 6) were the shared haplotypes (Figure 6). Haplotype 2 was the most abundant haplotype and distributed in five locations (Suihua city, Shuangyashan city, Qitaihe city, Qiqihar city, and Wuchang city). Haplotype 4 was found in Harbin city, Qiqihar city, Suihua city, and Shuangyashan city. Haplotype 1 was detected in Qiqihar city, Suihua city, and Shuangyashan city. Haplotype 6 was present in Qiqihar city, Suihua city, and Jiamusi city. Haplotype 5 was only found in two locations (Qiqihar city and Harbin city). The other five haplotypes were all private haplotypes. In addition, the haplotypes detected in Suihua city and Shuangyashan city were the most abundant and included five haplotypes, followed by Qiqihar city (four haplotypes), Harbin city (three haplotypes), Qitaihe city (three haplotypes), Wuchang city (one haplotype), and Jiamusi city (one haplotype).

## 3. Discussion

There are few records about maize sheath rot in the world. White et al. (1999; 2005) first recorded the occurrence of purple sheath blight disease on maize in the United States, which began to occur at the silking stage of maize, and only harmed the leaf sheath without infecting the leaves and stalks. It was considered that it may be caused by *Fusarium* spp. Purple sheath blight is common in the U.S. Corn Belt, but it does not result in yield loss. Since then, there have been almost no reports on maize sheath rot abroad. However, maize sheath rot was first recorded in 2008 in northeast China and has been reported to cause varying degrees of maize yield loss. In recent years, maize sheath rot has frequently occurred, affecting the yield of maize; for example, Li et al. [5] reported that sheath rot occurred in several maize fields of Liaoning, Jilin, and Heilongjiang provinces, causing an approximately 30% yield loss; Sun et al. [7] found sheath rot presented on approximately 40% of maize leaf sheaths in three 7 ha commercial fields in Heilongjiang province. Moreover, it has also been reported that serious infection with sheath rot could reduce the lodging resistance of maize [6,8].

Maize sheath rot occurs during the reproductive stage of maize, and the silking stage shows the strongest infection. The occurrence of maize sheath rot is closely related to many factors, including the maize variety, climate conditions, and the feeding of aphids [2]. A previous study showed that the disease incidence at the silking stage of maize was significantly higher than that of the tasseling stage, grain filling stage, and physiological maturity. The activities of defense enzymes at the silking stage of maize were also significantly higher than at the other stages. In addition, hybrid maize varieties were relatively more resistant than the inbred lines, and maize grown in a humid environment and a warm climate was more susceptible to *Fusarium* infection [35]. In the survey conducted in Heilongjiang province from 2021 to 2022 in this study, during the reproductive stage of maize, the disease incidence of maize sheath rot was 56.8–76.6% and the diversity of *Fusarium* species causing maize sheath rot gradually increased.

In the present study, 132 isolates belonging to 10 *Fusarium* species were identified as the causal agents of maize sheath rot in the main maize-growing areas of Heilongjiang province, and *F. subglutinans*, *F. temperatum*, *F. solani*, *F. sporotrichioides*, *F. tricinctum*, and *F. acuminatum* were first reported as the causal agents of maize sheath rot worldwide. Moreover, four *Fusarium* species—*F. verticillioides* (*n* = 66), *F. subglutinans* (*n* = 25), *F. temperatum* (*n* = 7), and *F. proliferatum* (*n* = 1), including 99 isolates—were classified into the *Fusarium fujikuroi* species complex (FFSC), of which *F. verticillioides* was the most prevalent fungal pathogen on maize and has been reported to cause serious stalk rot, ear rot, sheath rot, and leaf blight in maize [8,11,36,37]. *F. subglutinans* could cause seedling disease and ear rot in maize, resulting in serious loss of yield [38]. *F. temperatum* is a common maize pathogen which could produce mycotoxins and cause plant disease [39]. *F. proliferatum* is the major pathogen of maize sheath rot and was first reported in 2008 by Xu et al.; it could produce enniatins, beauvericin, fumonisins, fusaproliferin, fusaric acid, fusarins and moniliformin [40], contaminating mainly maize and maize products [41]. However, only one *F. proliferatum* isolate was obtained in this study, suggesting that the diversity and population structure of *Fusarium* species associated with maize sheath rot may have changed and the changes may be related to natural environmental conditions, maize varieties, tillage patterns, and other objective factors [42]. These results require us to pay attention to the genetic evolution and population composition of pathogens associated with plant diseases and reasonably select fungicides and biocontrol agents, as well as appropriate cultivars.

Moreover, different *Fusarium* species showed differentiated aggressiveness on maize sheath. The average disease indexes of *F. tricinctum*, *F. proliferatum*, and *F. asiaticum* were higher than those of the other *Fusarium* species, which was different from the previous conclusion; the average disease index of *F. proliferatum* was the highest, while the average disease index of *F. asiaticum* was relatively low [8], indicating that the disease index may be affected by the inoculation method [43]. This study is the first to clarify and compare the pathogenicity of *F. subglutinans*, *F. temperatum*, *F. solani*, *F. sporotrichioides*, *F. tricinctum*, and *F. acuminatum* associated with maize sheath rot in Heilongjiang province. These *Fusarium* species with different pathogenicities could also be used to evaluate the resistance of different maize varieties in future and will be helpful in breeding new resistant maize varieties [44].

In addition, the haplotypes of 99 FFSC isolates were identified, and 10 haplotypes were detected. Haplotypes 1, 2, 3, 7, 8, and 9 belonged to the *F. verticillioides* clade, and the haplotype network was radial with the center of haplotype 2, indicating that the population expanded rapidly after reaching the bottleneck [45]. Furthermore, haplotypes 4 and 5 belonged to the *F. subglutinans* clade, haplotype 6 belonged to the *F. temperatum* clade, and haplotype 10 belonged to the *F. proliferatum* clade. These members of the FFSC were assigned to different clades in the haplotype network, suggesting that the haplotype network could effectively distinguish *Fusarium* species in the complex, and also confirming our classification results. Moreover, the major haplotypes (Haps 1, 2, 4, and 6) presented in multiple locations of Heilongjiang province, and no significant correlation was found between geographic origin and haplotype distribution. Spores of the *Fusarium* species may be transmitted by wind, insects, or the trade of agricultural products [46,47,48,49,50], which might explain the wide geographic distribution of FFSC isolates in Heilongjiang province.

## 4. Materials and Methods

### 4.1. Sample Collection and Fusarium Isolation

A total of 98 symptomatic maize leaf sheath samples were collected from seven cities in the main maize-growing areas of Heilongjiang province during 2020–2021. The samples were kept in paper bags and stored in a 4 °C refrigerator. The maize leaf sheaths were cut into 0.5 cm × 0.5 cm pieces at the junction between diseased and healthy tissue prior to isolation. The leaf sheath pieces were soaked in 1% sodium hypochlorite for 1 min, rinsed twice with sterile distilled water, and placed on potato dextrose agar (PDA) plates supplemented with 100 μg/mL nalidixic acid sodium salt, for 3 days at 25 °C, for fungal isolation. All fungal colonies were picked out from the plates; however, only isolates identified as *Fusarium* species using morphological observation were selected for further study. Colony purification was carried out by picking a single spore using a stereomicroscope and culturing it on PDA at 25 °C for 7 days. In total, 132 *Fusarium* isolates were obtained (Appendix A) and preserved on PDA slants at 4 °C [51]. In addition, several other fungal colonies (including 4 *Didymella americana* isolates and 2 *Nigrospora musae* isolates) were also obtained in addition to these *Fusarium* isolates, but those fungi were all not pathogenic to maize leaf sheath.

### 4.2. Morphological Characterization

All isolates were inoculated on PDA plates and incubated at 25 °C in the dark for 7 days. Each isolate was assessed based on colony texture and colony color. In addition, these isolates were cultured on PDA at 25 °C for 14 days with a light/dark cycle of 8/16 h to observe the well-developed macroconidia. The macroconidia were evaluated using light microscopy (Zeiss Axiolab 5 equipped with an Axiocam 208 color industrial digital camera) [52].

### 4.3. DNA Extraction and Sequencing

Fungal DNA was extracted using 2% cetyltrimethylammonium bromide as Leslie and Summerell described [53]. The primers ITS1 (TCCGTAGGTGAACCTGCGG)/ITS4 (GCTGCGTTCTTCATCGATGC) and EF1-728F (CATCGAGAAGTTCGAGAAGG)/EF4-986R (TACTTGAAGGAACCCTTACC) were used to amplify the partial rDNA-ITS and TEF1-α genes, respectively [54,55]. The PCR products were purified and sequenced at Beijing Biootech Co. Ltd. The obtained gene sequences of 132 *Fusarium* isolates were aligned in GenBank and the *Fusarium* ID-database, and then, deposited into the NCBI GenBank.

### 4.4. Phylogenetic Analysis of Fusarium Isolates

In the present study, MEGA 7 software with default parameters was used for phylogenetic analysis, and the sequences were aligned based on the Clustal W algorithm. A phylogenetic tree based on the TEF-1α gene sequences was constructed using 1000 bootstrap repeated with the maximum likelihood (ML) method. *Alternaria junci-acuti* IRAN 3512C was selected as the outgroup.

### 4.5. Pathogenicity Tests

Forty-seven representative *Fusarium* isolates were selected for the pathogenicity test and the selection method and number are shown in Appendix A. All 47 *Fusarium* isolates were inoculated into 50 mL carboxymethyl cellulose (CMC) liquid medium (7.5 g sodium carboxymethyl cellulose, 0.5 g yeast extract, 2.5 g K_2_HPO_4_, and 0.25 g MgSO_4_·7H_2_O were added to 1000 mL water) and cultured for 3 days on a rotary shaker at 25 °C, 160 r.p.m. [56]. The conidia suspension was filtered and adjusted to 1 × 10^6^ conidia/mL. Healthy maize plants (var. Zhengdan 958; nine maize plants for each isolate) grown in the greenhouse at the heading stage and were selected for the pathogenicity test to fulfill Koch’s postulates. The maize leaf sheaths were surface-disinfected and wounded with a sterile needle (1mm diameter), and then, 10 μL of conidia suspension was injected into the maize sheaths. Three control maize leaf sheaths were inoculated with 10 μL sterile water. Symptoms were observed after 25 days post inoculation, and disease severity was scored according to the modified method described by Zhang et al. [57]. The experiment was repeated twice with three maize leaf sheaths per replication.

The disease index was assessed based on a 0–3 scale—0 (no obvious lesion), 1 (diameter of lesion between 0–0.3 cm), 2 (diameter of lesion between 0.3–0.5 cm), and 3 (diameter of lesion above 0.5 cm). The disease index (DI) was calculated by following the formula: DI = [100 × ∑ (n × corresponding DS)]/(N × 3), where n is the number of infected inoculation sheaths corresponding to each disease rating, and N is the total number of inoculation sheaths. SPSS software (v. 20.0; SPSS Inc., Wacker Drive, Chicago, IL, USA. IBM Corp., 2012. IBM) was used for statistical analysis using a least significant difference (LSD) test at a significance level of *p* < 0.05. All *Fusarium* isolates were re-isolated from the symptomatic maize leaf sheaths and identified based on morphological and molecular methods.

### 4.6. Nucleotide Diversity and Haplotype Assignment within Species

The ITS and TEF-1α gene sequences of 99 *Fusarium fujikuroi* species complex (FFSC) isolates were individually concatenated for haplotype analysis. PopART software version 1.7 (Population Analysis with Reticulate Trees is free, open-source population genetics software that was developed as part of the Allan Wilson Centre Imaging Evolution Initiative) was used for generating a genealogical network based on the concatenated sequences [58]. Haplotype identification based on parsimony probability computed for pairwise comparisons was carried out using a TCS network [59,60].

### 4.7. Data Analysis

Differences in the field survey, growth rate, and pathogenicity were analyzed using the Statistical Package for Social Sciences (SPSS) (v. 20.0; SPSS Inc., Wacker Drive, Chicago, Illinois.IBM Corp., 2012. IBM). An analysis of variance was performed using the general linear model, and the means were compared using Duncan’s New Multiple Range test in SPSS, with differences considered significant at *p* ≤ 0.05.

## 5. Conclusions

In conclusion, 10 *Fusarium* species were isolated from seven major maize-growing areas in Heilongjiang province, China. *F. verticillioides* was the dominant species. *F. subglutinans*, *F. temperatum*, *F. solani*, *F. sporotrichioides*, *F. tricinctum*, and *F. acuminatum* were first reported as the casual agents of maize sheath rot. Clarifying the population structure and pathogenicity of *Fusarium* spp. associated with maize sheath rot in Heilongjiang province will be helpful in studying resistant maize varieties, and in selecting the control agents and providing guidelines for the comprehensive control of maize disease in Heilongjiang province.

## Figures and Tables

**Figure 1 ijms-23-10821-f001:**
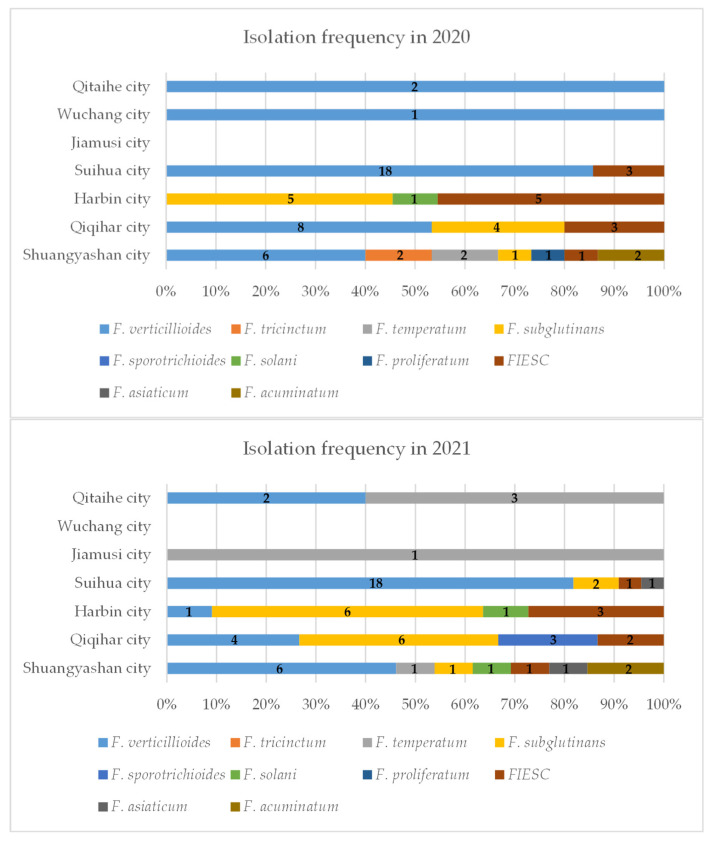
Isolation frequency of Fusarium isolates collected from 7 locations of Heilongjiang Province in 2020 and 2021, respectively.

**Figure 2 ijms-23-10821-f002:**
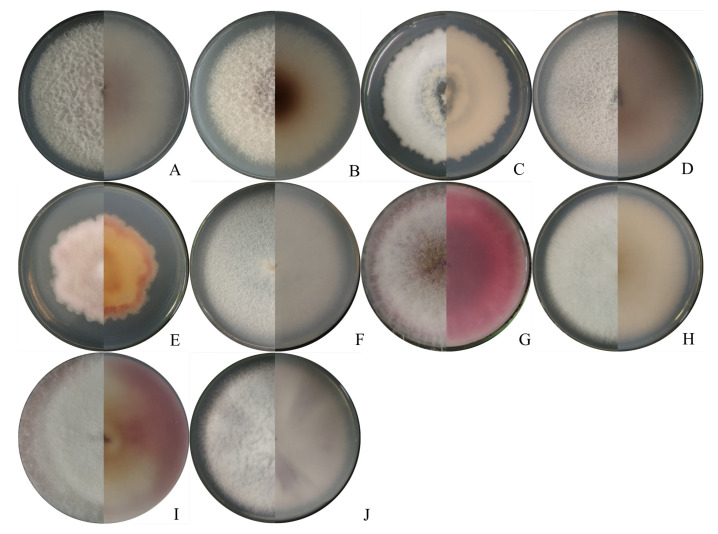
Colony appearance of representative isolates of 10 *Fusarium* species. (**A**–**J**) Colonies of the representative isolates of *F. verticillioides*, *F. subglutinans*, *Fusarium incarnatum*-*equiseti* species complex, *F. temperatum*, *F. acuminatum*, *F. solani*, *F. sporotrichioides*, *F. tricinctum*, *F. asiaticum*, and *F. proliferatum*, respectively.

**Figure 3 ijms-23-10821-f003:**
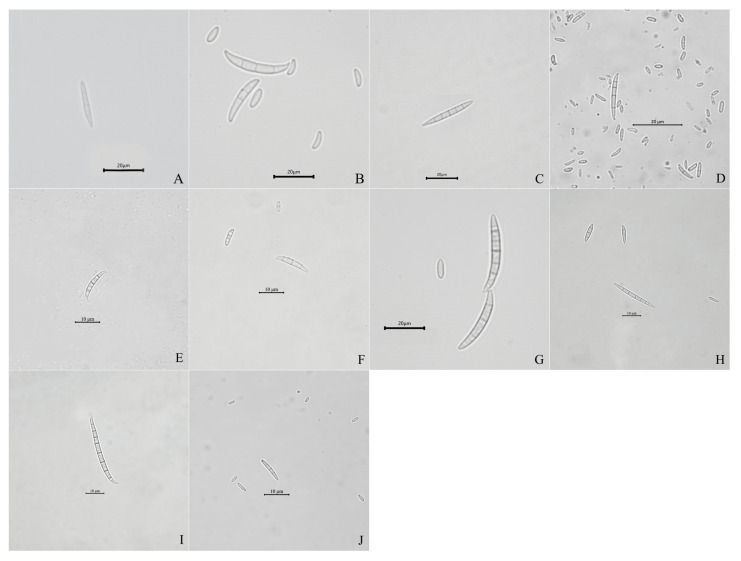
Conidia morphology of representative isolates of 10 *Fusarium* species. (**A**–**J**) Macroconidia or microconidia of representative isolates of *F. verticillioides*, *F. subglutinans*, *Fusarium incarnatum*-*equiseti* species complex, *F. temperatum*, *F. acuminatum*, *F. solani*, *F. sporotrichioides*, *F. tricinctum*, *F. asiaticum*, and *F. proliferatum*, respectively.

**Figure 4 ijms-23-10821-f004:**
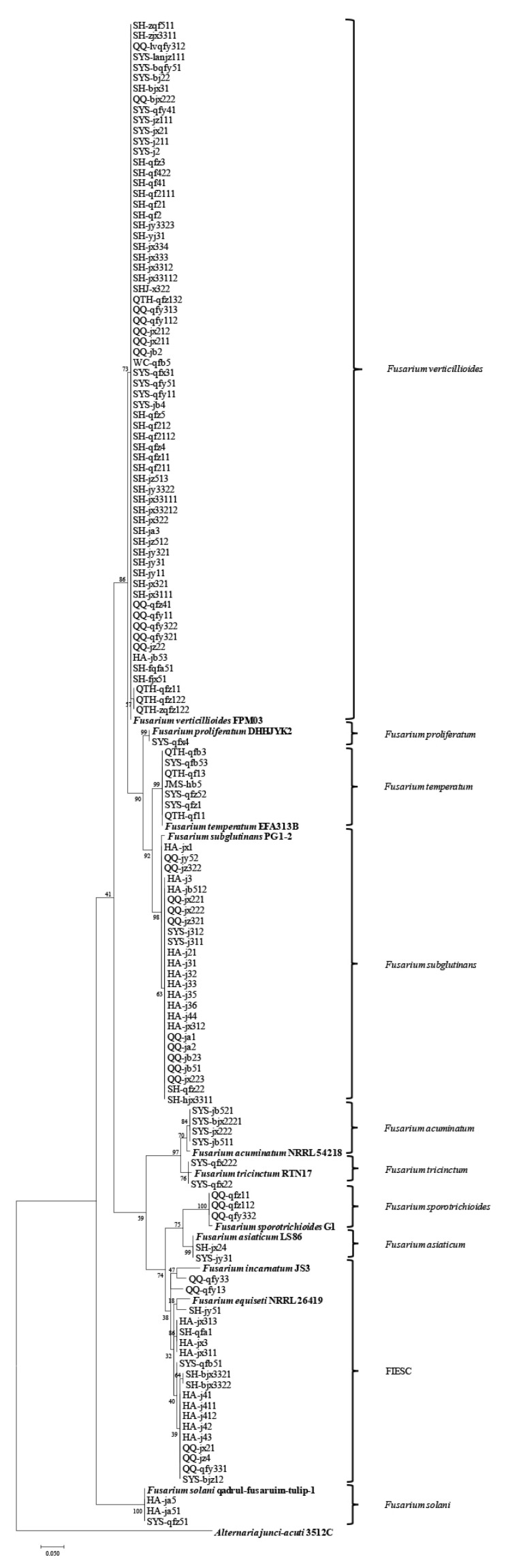
Phylogenetic tree obtained from maximum likelihood analysis based on the TEF-1α gene sequences.

**Figure 5 ijms-23-10821-f005:**
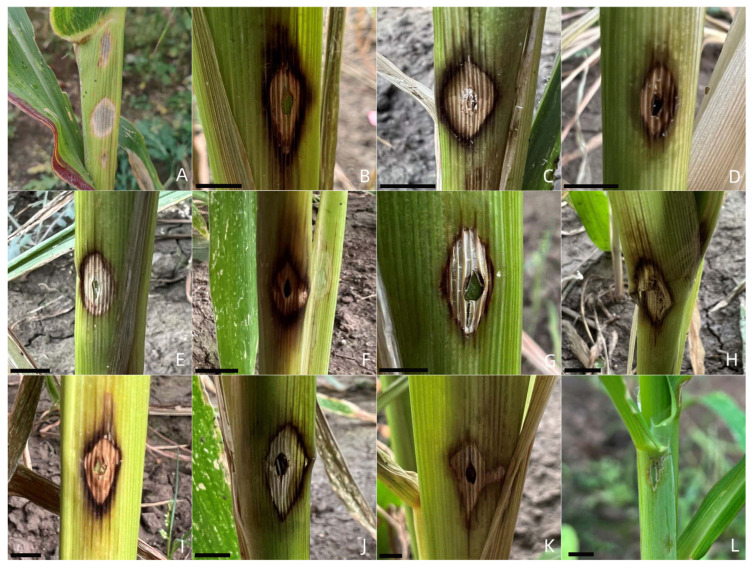
Inoculation of maize leaf sheaths with different *Fusarium* species (var. Zhengdan 958). (**A**) Maize sheath rot symptoms observed in the field; (**B**–**K**) typical symptoms on maize sheaths caused by *F. verticillioides*, *F. subglutinans*, *Fusarium incarnatum-equiseti* species complex, *F. temperatum*, *F. acuminatum*, *F. solani*, *F. sporotrichioides*, *F. tricinctum*, *F. asiaticum*, and *F. proliferatum*, respectively; (**L**) CK. Bar = 5 mm.

**Figure 6 ijms-23-10821-f006:**
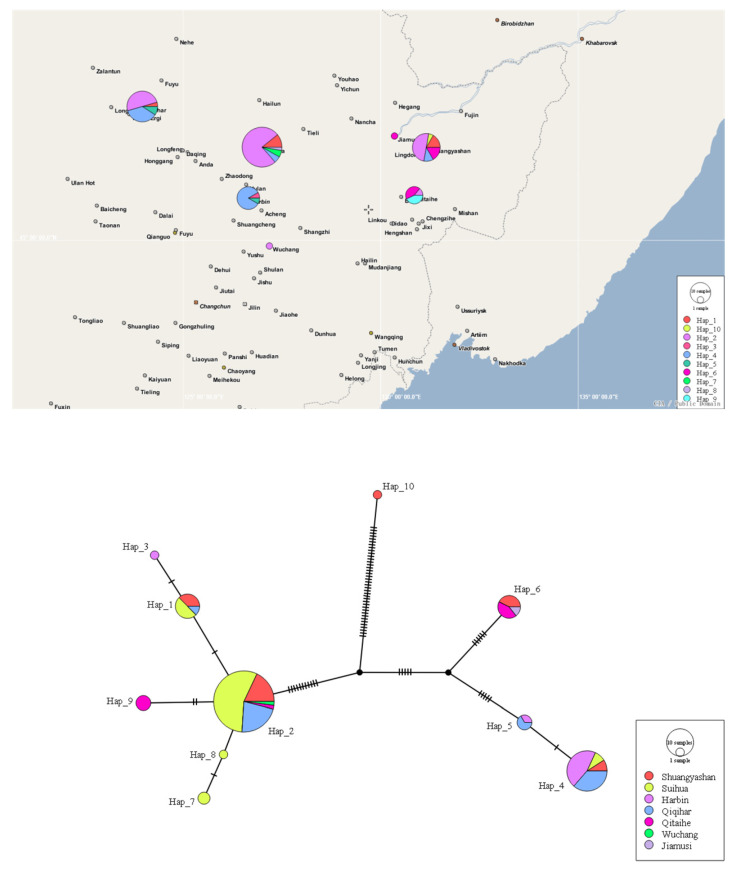
Haplotype distribution and TCS analyses based on the concatenated sequences of ITS and TEF-1α genes of 99 FFSC isolates obtained from different locations of Heilongjiang province, China. Each haplotype is represented by a circle, the size of which is proportional to the haplotype frequency.

**Table 1 ijms-23-10821-t001:** The morphological characteristics of *Fusarium* species obtained in this study.

Groups	Colonies Appearance	Conidia			
		Length (μm)	Width (μm)	Septum	Shape
*F. verticillioides*	White to greyish-purple mycelia	16.6–23.4	2.3–3.4	3–4	Slightly falcate to almost straight
*F. subglutinans*	Yellow mycelia, and then, turned lavender	13.6–21.1	1.6–2.3	3–5	Slightly falcate
FIESC	White to light beige with loosely floccose mycelia	16.0–18.0	1.5–3	3-6	Falcate
*F. temperatum*	White to peach-colored mycelia	13.6–23.1	1.6–3.3	3-5	Falcate
*F. acuminatum*	Abundant and partly carmine mycelia	13.2–24.2	1.9–3.7	3-5	Slender, equilaterally curved
*F. solani*	Dense and white mycelia	16.0–26.0	1.5–2.8	2-4	Sickle-shaped
*F. sporotrichioides*	White mycelia, and then, turned pink	12.5–24.3	3.1–4.2	3-5	Sickle-shaped
*F. tricinctum*	White to canary yellow, and dense mycelia	12.5–26.0	1.5–3.0	3-5	Falcate
*F. asiaticum*	White mycelia with pink pigmentation	13.0–36.0	1.6–3.8	4-8	Falcate
*F. proliferatum*	White mycelia with dark violet pigmentation	13.6–22.3	3.1–4.2	3-4	Slender

**Table 2 ijms-23-10821-t002:** Disease index and severity degree of maize leaves inoculated with 10 *Fusarium* species.

Fusarium Species	Disease Index ^a^
F. tricinctum	88.9–100 (94.4 ± 6.4) a
F. proliferatum	88.9–100 (94.4 ± 5.6) a
F. asiaticum	33.3–100 (73.6 ± 20.5) ab
F. acuminatum	33.3–88.9 (68.1 ± 17.3) ab
F. solani	22.2–100 (61.1 ± 39.0) ab
F. sporotrichioides	33.3–100 (58.3 ± 29.2) b
F. temperatum	44.4–77.8 (58.3 ± 16.7) b
F. verticillioides	33.3–100 (57.4 ± 26.7) b
F. subglutinans	22.2–100 (53.7 ± 32.5) b
FIESC	11.1–77.8 (42.3 ± 22.6) b

^a^ Maize sheaths were surface-disinfected and wounded with a sterile needle, and then 10 μL of conidia suspension (1 × 10^6^ spores/mL) was injected into maize sheaths (var. Zhengdan 958). Disease severity (DS) was scored after 25 days incubation at 25 °C and 90% relative humidity using a 0–3 scale. Numbers outside the parentheses are the range of disease incidence and disease index on sheaths of maize inoculated with corresponding *Fusarium* species. Values in parentheses are the mean ± standard deviation based on the data of each tested *Fusarium* isolate of the corresponding species. Values followed by different lowercase letters within a column are significantly different according to the least significant difference test (*p* < 0.05).

## Data Availability

Sequences have been deposited in GenBank. The data presented in this study are openly available in NCBI. Publicly available datasets were analyzed in this study. These data can be found here: https://www.ncbi.nlm.nih.gov/.

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
