# Peer review of "Identification, Pathogenicity, and Genetic Diversity of Fusarium spp. Associated with Maize Sheath Rot in Heilongjiang Province, China"

_ijms, 2022, doi:10.3390/ijms231810821_

Round 1

Reviewer 1 Report

The manuscript describes identification and characterization of Fusarium sp. associated with corn rots in local province in China in 2020 and 2021. The results show that Fusarium species associated with corn rots in the area studied were more diverse than previously reported, and many new species were identified for the first time in the area.    

Author Response

Thank you very much.

Reviewer 2 Report

The manuscript is well written and has the merits. Although it is a quite traditional study but the authors successfully incorporated modern techniques to signify the value of the work. 

However, I have some queries/comments which should be addressed before further consideration of the manuscript

There is a need to include a well defined experiment setup in the methodology section and information regarding the statistical analysis of the data. 

The authors should make a table highlighting the morphological characteristics of the fusairum species.

Phylogenetic tree is not readable.

The discussion section should also include the information about the economic impact of findings. How the findings will be helpful in developing suitable management strategies including genetic resistance and etc. And how the knowledge obtained will help in resistance break down in plants. The discussion need to be elaborated.

Conclusion is not highlighting the impact of the outcome of the study and future perspectives, the authors need to work on that section as well.

Author Response

Thank you very much for your valuable suggestions.

  1. There is a need to include a well defined experiment setup in the methodology section and information regarding the statistical analysis of the data.

Thank you very much for your valuable advice. We have added a data analysis. (Please see lines 145-149).

  1. The authors should make a table highlighting the morphological characteristics of the Fusarium

Thank you for your valuable suggestion. We have added Table 1 about the morphological characteristics of Fusarium species. (Please see Table 1).

  1. Phylogenetic tree is not readable.

Thank you very much for your valuable advice. We have revised Fig. 4.

  1. The discussion section should also include the information about the economic impact of findings. How the findings will be helpful in developing suitable management strategies including genetic resistance and etc. And how the knowledge obtained will help in resistance break down in plants. The discussion needs to be elaborated.

Thank you for your valuable suggestion. We have revised. (Please see lines 275-298, 314-318, 326-328, and 346-348).

  1. Conclusion is not highlighting the impact of the outcome of the study and future perspectives; the authors need to work on that section as well.

Thank you for your valuable suggestion. We have added. (Please see line 346-348).

Reviewer 3 Report

The paper represents a report on maize sheath rot and pathogens responsible for this disease found in China. Authors performed sampling and morphological / genetic analysis of species diversity. Overall the study deserves attention and has potential for applications, but the paper can be optimized with consideration of suggestions for improvement:

Line 12 – the word “maize” is incidentally bold

Line 36-37 – I suggest to add a sentence about the maize importance worldwide as a food crop

Line 50 – were → was

Line 162 - double space after the word “Province”

Line 156 - I would move the Suppl Fig S1 and S2 into the results part / main text to combine them with the morphology description and figure references in the text for better understanding and clearness of results

Lines 167-170 - reference to a figure needed

Lines 175-177 - reference to a figure for better reading

Lines 185-187 - reference to a figure / please check in text

Lines 211-212 - difficult to read: maybe move into supplementary? / improve the resolution/quality

Line 267 - reference / citation is needed

Line 287 – I suggest to add the information about the importance of the study in the worldwide scope

Line 306 - I suggest to add a sentence about the overall importance / future perspectives for applications that arise from the current study

Author Response

Thank you very much for your valuable suggestions.

  1. the word “maize” is incidentally bold

Thank you very much. We have revised. (Please see line 12)

  1. Line 36-37 – I suggest to add a sentence about the maize importance worldwide as a food crop

Thank you for your suggestion. We have added, please see lines 37-39.

  1. Line 50 - were → was

Thank you very much. We have revised. (Please see line 47).

  1. Line 162 - double space after the word “Province”

Thank you very much. We have revised. (Please see line 163).

  1. Line 156 - I would move the Suppl Fig S1 and S2 into the results part / main text to combine them with the morphology description and figure references in the text for better understanding and clearness of results;

Lines 167-170 - reference to a figure needed

Lines 175-177 - reference to a figure for better reading

Lines 185-187 - reference to a figure / please check in text

Thank you very much. We have moved the Suppl Fig S1 and S2 into the results part as Fig. 1 and Fig. 2. (Please see Fig. 1 and 2).

  1. Lines 211-212 - difficult to read: maybe move into supplementary? / improve the resolution/quality.

Thank you very much for your valuable advice. We have improved the resolution/quality of Fig. 4. (Please see Fig. 4).

  1. Line 267 - reference / citation is needed.

Thank you very much for your valuable advice. The reference is 48, please see lines 308 and 480-481.

  1. Line 287 – I suggest to add the information about the importance of the study in the worldwide scope.

Thank you very much for your valuable advice. We have added. (Please see lines 275-287).

  1. Line 306 - I suggest to add a sentence about the overall importance / future perspectives for applications that arise from the current study.

Thank you very much for your valuable advice. We have added. (Please see lines 346-348).

Round 2

Reviewer 2 Report

The changes made are acceptable.